# Revision Surgery for Achilles Tendon Rupture: A Comprehensive Review of Treatment Options, Outcomes, and Complications and the Role of Artificial Intelligence

**DOI:** 10.3390/medicina61091684

**Published:** 2025-09-17

**Authors:** Elena Delmastro, Stefano Colace, Umile Giuseppe Longo, Pieter D’Hooghe, Alberto Marangon, Olimpio Galasso, Giorgio Gasparini, Michele Mercurio

**Affiliations:** 1Orthopaedic and Trauma Department, Università Vita-Salute San Raffaele, 20132 Milano, Italy; delmastro.elena@gmail.com; 2Department of Orthopaedic and Trauma Surgery, “Magna Græcia” University, “Renato Dulbecco” University Hospital, 88100 Catanzaro, Italy; gasparini@unicz.it (G.G.); michele.mercurio@unicz.it (M.M.); 3Fondazione Policlinico Universitario Campus Bio-Medico, 00128 Roma, Italy; g.longo@policlinicocampus.it; 4Research Unit of Orthopaedic and Trauma Surgery, Department of Medicine and Surgery, Università Campus Bio-Medico di Roma, 00128 Rome, Italy; 5Department of Orthopedic Surgery, Aspetar Orthopaedic and Sports Medicine Hospital, Doha 29222, Qatar; pieter.dhooghe@aspetar.com; 6Clinic San Francesco, 37127 Verona, Italy; alberto.marangon@me.com; 7Department of Medicine, Surgery and Dentistry, University of Salerno, Baronissi, 84081 Salerno, Italy; ogalasso@unisa.it; 8Research Center on Musculoskeletal Health, MusculoSkeletalHealth@UMG, Magna Graecia University, 88100 Catanzaro, Italy

**Keywords:** Achilles tendon rupture, allograft, artificial intelligence, rehabilitation, revision surgery, tendon transfer, treatment algorithms, turndown flaps

## Abstract

*Background and Objectives*: Achilles tendon ruptures in middle-aged individuals with systemic comorbidities represent a growing clinical challenge. Revision surgery, indicated in cases of tendon re-rupture, remains technically demanding and lacks standardized treatment protocols. This comprehensive review aimed to summarize current evidence regarding indications, outcomes, and complications associated with the most commonly employed revision techniques and explores the potential of artificial intelligence (AI) in improving management and outcomes. *Materials and Methods*: A literature review was performed in accordance with PRISMA guidelines. The PubMed, MEDLINE, and Cochrane Central databases were used to search keywords. We included articles (1) reporting indications, outcomes, and/or complications of revision surgery for Achilles tendon rupture; (2) reporting a minimum mean follow-up of >12 months; and (3) written in English. Six studies met the inclusion criteria, with a total of 3250 patients analyzed. A methodological quality assessment using the Modified Newcastle–Ottawa Quality Assessment Scale was performed, and all articles were found to be of high quality. *Results*: Surgical strategies were stratified based on defect size: <2 cm: end-to-end anastomosis; 2–5 cm: V-Y myotendinous lengthening, often combined with tendon transfer; and >5 cm: fascial turndown flaps, autografts (e.g., semitendinosus), or allografts. Tendon transfers showed satisfactory functional outcomes but varied in complication rates. Allografts offered reduced donor site morbidity. The use of AI and wearable sensors has demonstrated potential in preoperative planning, complication prediction, and real-time rehabilitation monitoring. *Conclusions*: Achilles tendon revision surgery requires a patient-specific, defect-oriented approach. Combined surgical techniques are often necessary for large or non-viable lesions. The integration of AI represents a promising advancement in enhancing surgical decision-making, optimizing rehabilitation, and improving long-term clinical outcomes.

## 1. Introduction

Achilles tendon injuries are increasingly common, particularly among middle-aged and older adults, who are also more likely to have systemic comorbidities that elevate the risk of post-surgical complications and revision surgery [1,2,3,4,5]. The overall complication rate following primary Achilles tendon repair is approximately 2.7%, with a reported two-year revision rate of 4.3% [6]. However, the specific risk factors for repair failure remain controversial. Various cohort studies have cited factors such as hypertension, obesity, younger age, underlying tendinopathy, and postoperative complications as potential contributors. Metabolic disorders, including diabetes mellitus, hyperuricemia, dyslipidemia, and obesity, are associated with Achilles tendon rupture. These conditions induce tendon degeneration through collagen alterations, microvascular compromise, and chronic low-grade inflammation. Advanced glycation end-products and lipid accumulation reduce tendon elasticity and structural integrity. Oxidative stress further impairs tissue resilience. Collectively, these factors increase the risk of rupture and may delay tendon healing in affected patients [6,7,8,9]. The socio-economic burden of revision surgery is also noteworthy. While the average five-year cost for primary Achilles tendon surgery is estimated at $17,307, the need for revision surgery can add approximately $6776.40 to overall expenditures [6]. Revision procedures present unique challenges for orthopedic surgeons, often involving large tendon defects, tissue degeneration, and poor local vascularity [8]. In recent years, attention has increasingly focused on optimizing surgical technique selection through detailed preoperative evaluations that consider injury patterns, patient-specific factors, and their influence on outcomes. Given the complex and evolving nature of revision surgery for Achilles tendon ruptures, this comprehensive review aims to summarize current evidence regarding indications, outcomes, and complications associated with the most employed revision techniques. Indeed, considering the many surgical procedures proposed in the literature and the possible technical variations, it is not easy for the surgeon to choose which technique to use based on the specific type of lesion and pathological anatomy and what to expect in the postoperative period. This review primarily aimed to define the current management of Achilles tendon revision surgery. Secondarily, the role of artificial intelligence in supporting orthopedic surgeons in diagnosis, treatment, and rehabilitation was also explored. This study may help to guide evidence-based decision-making, ultimately improving patient outcomes and awareness of the challenging management of revision surgery for Achilles tendon rupture.

## 2. Materials and Methods

### 2.1. Search Strategy

According to the Preferred Reporting Items for Systematic Reviews and Meta-Analyses (PRISMA) statement, a comprehensive review of the published literature was conducted and reported. The PubMed, MEDLINE, and Cochrane Central databases were searched in September 2024 for the literature review with no lower date limit. The search terms “Achilles” AND “tendon” AND “revision” AND “surgery” were used in different combinations to retrieve relevant articles. The articles were selected based on the following PICO model: (P) patients with Achilles tendon rupture; (I) patient underwent Achilles tendon revision surgery; (C) regardless of the presence or absence of comparator or control groups, all studies were eligible; and (O) patients assessed for functional outcomes and complications.

Two authors (ED and SC) independently reviewed the titles and abstracts and contacted a third senior author (MM) if there were significant discrepancies. The references list of each article and the available gray literature at our institution were screened for potential additional articles.

### 2.2. Inclusion Criteria and Quality Assessment

We included articles (1) reporting indications, outcomes and/or complications of revision surgery for Achilles tendon rupture; (2) reporting a minimum mean follow-up of >12 months; and (3) written in English. Other reviews, case reports, cadaveric or biomechanical studies, technical notes, editorials, letters to the editor, and expert opinions were excluded from the analysis but considered for the Discussion Section [10]. A methodological quality assessment using the Modified Newcastle–Ottawa Quality Assessment Scale was independently performed by 2 authors (ED and SC). Disagreements were resolved by consultation with a senior reviewer (MM). The details of the quality assessment are shown in Table 1. Substantial interobserver agreement (Cohen kappa coefficients ranging between 0.59 and 0.74) is reported.

## 3. Results

In total, the initial search identified 205 relevant articles; 198 were excluded. This resulted in six studies that were eligible for this review (Figure 1).

The included studies were published from 2013 to 2024. Two studies [15,16] were conducted in Italy, one [11] in France, one [7] in South Korea, and two [6,14] in the United States. A total of 3250 patients were initially identified (Table 2).

### 3.1. Treatment Algorithms

Myerson (1999) and Buda (2017) proposed treatment algorithms for neglected Achilles tendon ruptures, which can also be adapted to Achilles tendon re-ruptures based on the size of the non-viable tendon gap [17,18]. However, it is important to note that chronic ruptures and re-ruptures are not identical entities and may differ in their underlying pathology and healing potential. The scarcity of literature specifically addressing Achilles tendon re-ruptures poses a significant challenge in establishing a standardized treatment protocol.

According to the Myerson and Buda algorithms:
Defects measuring 1–2 cm can typically be treated with a simple end-to-end anastomosis. Buda also suggests considering augmentation or tendon transfer, if necessary, with the plantaris tendon being his preferred option in such cases.Defects between 2–5 cm are best managed with V-Y myotendinous lengthening in combination with end-to-end repair. When tendon quality is compromised, flexor hallux longus (FHL) transfer may be warranted. In these instances, Buda favors a turndown flap if feasible.Defects greater than 5 cm require more extensive reconstruction. Myerson recommends combining turndown flaps and FHL transfer with other techniques as needed. Buda proposes a more aggressive approach, advocating for one or two tendon transfers in addition to V-Y lengthening, with optional use of allografts when necessary.

For large or complex tendon gaps, both authors acknowledge the utility of autografts or allografts as adjuncts to reconstruction. Myerson particularly emphasizes their role in bridging extensive deficits when local tissue is insufficient [9,17,19].

It is critical to recognize that the choice of surgical intervention in Achilles tendon revision is heavily influenced by patient-specific clinical factors. Patients presenting with uncontrolled diabetes mellitus, recurrent infections, or rheumatologic disorders necessitating biologic therapies—which may impair tendon regenerative capacity—are frequently more appropriate candidates for tendon replacement procedures. Careful consideration of these factors is essential to optimize functional outcomes, enhance tissue healing, and minimize the risk of postoperative complications. In this context, artificial intelligence can support clinical decision-making by analyzing patient-specific data and historical outcomes to recommend the most suitable surgical approach for each individual.

Figure 2 presents an illustrative diagram with the techniques used and the indications according to the type of lesion and tendon quality.

#### 3.1.1. End-to-End Anastomosis

End-to-end anastomosis is typically indicated in cases involving small tendon defects (<2 cm) and good tendon quality. Various suture techniques—such as Krackow, Kessler, and Bunnell—are commonly employed, with suture materials ranging from polypropylene to ultra-high-molecular-weight polyethylene. However, the literature lacks consensus on the optimal technique [18,20]. Notably, the use of nonabsorbable braided sutures has been associated with higher rates of postoperative wound complications [21].

While the surgical principles are consistent with those used in primary repairs, revision cases require thorough debridement of degenerative tissue. Small amounts of scar tissue may be interposed if necessary [22]. Percutaneous techniques are rarely employed in revisions due to the need for direct tendon assessment and the necessity to release adhesions, which are frequently encountered in these cases. Postoperative care is crucial to minimize the risk of new soft tissue adhesions.

#### 3.1.2. V-Y Myotendinous Lengthening

V-Y myotendinous lengthening is the most used lengthening technique in Achilles tendon re-ruptures [23]. It involves a V-shaped incision at the proximal tendon near the musculotendinous junction, with the arms of the V measuring 1.5 to 2 times the tendon gap. The tendon stumps are approximated and sutured, typically using Krackow sutures, and the proximal incision is closed in a Y-shaped manner [24].

This technique offers the benefit of preserving push-off strength and reducing adhesion formation compared to turndown flaps [25]. However, it is rarely sufficient as a standalone procedure due to reports of decreased muscle power postoperatively [26,27]. Endoscopic-assisted modifications, such as that described by Xu et al., offer potential benefits in patients with wound healing concerns [28].

Other proximal lengthening techniques include the Baumann method and gastrocnemius recession (Stryer technique). The Vulpius technique is rarely applicable due to the typical location of Achilles tendon ruptures.

#### 3.1.3. Fascial Turndown

Fascial turndown, first introduced by Bosworth, is indicated for large tendon defects [29]. Variants involve creating a longitudinal strip from the central Achilles tendon, maintaining distal attachment, and weaving it through the proximal and distal tendon stumps using interrupted sutures [11,30]. Flap lengths up to 20 cm can be achieved.

A drawback of this technique is increased soft tissue bulk, potentially leading to wound dehiscence and discomfort. Combining fascial turndown with a V-Y lengthening flap can optimize outcomes [31,32]. A less invasive variation, described by Rammelt et al., may be beneficial for patients with fragile skin [33].

#### 3.1.4. Tendon Transfer

Flexor Hallucis Longus (FHL) is the most widely used tendon in Achilles revision surgery [34]. It is anatomically in-phase with the Achilles tendon, has robust plantarflexion strength, and improves vascularity at the repair site [35]. The tendon can be harvested via single or double incisions and fixed using interference screws, cortical buttons, or both [36,37,38,39,40,41]. FHL transfers are also more cost-effective than conservative management [42,43]. Limitations include potential hallux dysfunction and rigidity [44]. It could also be performed with an endoscopic approach. It is particularly promising and reliable, as it allows the procedure to be performed minimally invasively.

The plantaris tendon, when present, is easily harvested and functions harmoniously with the Achilles tendon. However, it is often scarred within the injured tendon, complicating harvest [45,46,47].

Peroneus Brevis (PB) tendon transfer, described by Pérez Teuffer, involves routing the PB tendon from the lateral to medial direction and then proximally toward the Achilles tendon [48,49]. Benefits include autologous tissue use, reduced infection risk, and procedural simplicity. However, it can decrease dorsiflexion and is contraindicated in cases of ankle instability or varus hindfoot deformity [50,51].

Peroneus Longus (PL) is used when other tendons are unavailable. It offers greater length and is fixed with a biotenodesis screw through a calcaneal tunnel [52].

Flexor Digitorum Longus (FDL), though less commonly used, offers a viable alternative. The procedure is technically demanding due to proximity to the neurovascular bundle [53].

#### 3.1.5. Autografts

Autografts, including semitendinosus-gracilis (ST-GR) and quadriceps tendon, offer solutions without compromising local tendon function. ST-GR grafts are harvested in supine position and show favorable outcomes [15,16]. The quadriceps tendon, although limited in length, offers strong fixation via its bony insertion [12,54].

In complex cases, options like an anterolateral thigh flap with vascularized fascia lata have demonstrated excellent functional recovery [13]. Comparative biomechanical studies have shown superior tensile strength of semitendinosus allografts over fascial turndown techniques [32].

#### 3.1.6. Allografts

Allografts—most commonly Achilles tendon with bone block—are a viable choice in patients for whom autografts are contraindicated [55,56]. They offer reduced surgical time and no donor site morbidity but carry small risks of disease transmission and immune response [57,58,59,60]. Functional outcomes are generally favorable, with minimal complications such as wound healing delays and heterotopic ossification [57]. While the Achilles tendon is the most commonly used allograft, alternative options, such as peroneus brevis, have been reported [61]. 

#### 3.1.7. Synthetic Materials

Synthetic grafts like LARS and polyester tapes have been utilized in chronic cases. However, they carry a relatively high complication rate (~20%), most commonly superficial wound infections. A small proportion of patients may require further surgery [62,63,64,65].

#### 3.1.8. Xenograft

Xenografts have demonstrated promising biomechanical performance when augmenting Krackow repair [66]. They may also be used alongside allografts with encouraging outcomes [67]. Nonetheless, their long-term biological integration remains under investigation.

## 4. Discussion

### 4.1. End-to-End Anastomosis

End-to-end anastomoses are indicated for small defects (<2 cm). Various suture techniques are available, but the use of non-absorbable sutures has been associated with post-operative complications. Regarding new suture constructs for end-to-end repairs, White and colleagues found no significant difference in construct elongation when comparing the traditional Krackow stitch with premanufactured locking loop stitch for soft tissue fixation. Overall, consideration of tendon quality, apposition feasibility, and reapproximating technique must all be scrutinized for proper revisional repair. Complications included surgical site infections, re-rupture, scar tissue formation and adhesions, tendon weakness or degeneration, chronic pain, compartment syndrome, nerve injury, delayed union or non-union, altered biomechanics, deep vein thrombosis and pulmonary embolism [68].

### 4.2. V-Y Myotendinous Lengthening

Lengthening techniques (V-Y and others) are used for larger defects, with the aim of avoiding adhesions and maintaining push-off strength. The V-Y technique is the most frequently used for Achilles tendon re-injuries but requires other associated surgical techniques to obtain a sufficient lengthening. Nonetheless, several studies have cited disadvantages of isolated V-Y advancements. Us and colleagues noted a 22% decrease in peak torque when using isolated V-Y lengthening. Similarly, Kissel and colleagues revealed a 30% decrease in functional power with the same procedure. Elais and colleagues in 2007 reiterated these concerns and recommended augmenting such a procedure with an FHL transfer to improve plantarflexory power [69,70,71].

Complications, in addition to those listed above for the other technique, include muscle weakness and dysfunction: the lengthening procedure, while aiming to relieve tension on the Achilles tendon, can lead to reduced strength in the calf muscles. This weakness may impair the patient’s ability to perform activities that require propulsion from the calf muscles, such as walking, running, or climbing stairs. Over-lengthening and loss of ankle plantarflexion was also reported: an excessive lengthening of the gastrocnemius–soleus complex can result in an abnormal loss of ankle plantarflexion, which is crucial for normal gait and mobility. The excessive reduction in calf muscle tension can affect the push-off phase during walking and running, leading to an inefficient and potentially altered gait pattern. Lengthening the gastrocnemius and soleus muscles may alter the biomechanical forces acting on the ankle joint. In some cases, this alteration can result in joint instability, particularly during dynamic movements. The reduced muscle tension can decrease the support for the subtalar joint, leading to issues with balance and control. In severe cases, this could contribute to chronic ankle instability and an increased risk of sprains or other injuries [48,72].

### 4.3. Fascial Turndown

Fascial turndown is used for severe tendon losses. While effective, this technique can cause discomfort due to increased thickness in the repaired area. Because of the increase in soft tissue volume at the repair site, complications involving wound dehiscence have been reported. Khiami and colleagues in 2013 proposed a modified technique of implementing a free sural triceps aponeurosis graft harvested proximally and transposed between the diseased tendon after performing a Z-plasty. Their technique not only reduced soft tissue thickness and complication risks but also allowed for length adjustments to be made during repair. The Mean American Orthopaedic Foot and Ankle Society (AOFAS) score was 96, with 12 of 16 patients returning to sport. MRI evaluation at 1 year post-operatively revealed homogeneous tendon and graft integration along with an increase in tendon size [73].

### 4.4. Tendon Transfer

The FHL is the most used for replacement, while other tendons such as the PB and PL are selected based on availability and patient conditions. Tendon transfers are effective but carry inherent risks such as loss of function (i.e., hallux rigidus in FHL, loss of stabilization during inversion in PB, loss of plantarflexion in PL and FDL).

Beyond clinical outcomes, a study by Hahn and colleagues examined tendon incorporation of the FHL on MRI at 3.8-year follow-up after transfer. Twelve of 13 tendons showed a significant degree of incorporation, and FHL muscle belly showed an average increase in size of 17%. The occurrence of both tendon incorporation and muscle belly hypertrophy revealed a functional adaptive ability of the FHL tendon regarding transfer to the Achilles tendon [74]. FHL and PB are usually the first choice, except when it is necessary to avoid functional impairment of the hallux in FHL and in case of cavo varus foot in PB. A recent meta-analysis reported that FHL and PB tendon transfer for chronic Achilles tendon ruptures resulted in favorable clinical outcomes and a reliable return to daily activities and sports [36]. The authors also compared the return to sport in FHL and in PB tendon transfer in chronic Achilles tendon injuries; while the PB group demonstrated a slower return to sport, a higher percentage of these patients eventually resumed sports activities compared to those who underwent FHL transfer [36]. Cesar Netto showed good results with this technique, with complication rates like other types of transfers (weakness of plantarflexion, infections) [75]. Maffulli et al. showed in a systematic review that the rate of complications following the use of FHL transfers is 14.8%. Of the 338 patients included, the major complications were one deep vein thrombosis, four deep infections, and one re-rupture. The endoscopic approach for FHL tendon transfer offers the potential for reduced soft-tissue disruption, improved visualization of the tendon and surrounding structures, and faster postoperative recovery [76]. Although the plantaris tendon is considered the easiest to harvest, pathophysiologic changes in a chronic or revisional Achilles tendon can often provide a challenge for revision surgery for the Achilles tendon surgeons. This tendon is usually heavily incorporated within the diseased Achilles tendon and may be difficult to identify and dissect away. However, if identified, it has the advantage of being harmoniously functional with the Achilles tendon. It also provides substantial autograft potential to reconstructive surgeons. PL can be a choice in case FHL was already used in previous surgery. Wang and colleagues in 2009 harvested the PL through several stab incisions and fixated the tendon into the plantar calcaneus with a cortical button. At a 2.5-year follow-up, MRI revealed maintenance of the PL within the calcaneal osseous tunnel and anatomic alignment of the Achilles tendon. No decrease in plantarflexion or hindfoot eversion was evident after active rehabilitation [77]. FDL can be another option even if the harvest can be more demanding due to the proximity of the neurovascular bundle. Complications regarding use of tendon transfers for Achilles tendon re-ruptures are infection, rejection or failure or rupture of the transferred tendon, functional problems, adhesions or scaring, nerve damage, difficulty in rehabilitation, persistent pain and disappointing functional outcomes

### 4.5. Autografts

Autografts such as semitendinosus and gracilis autografts are promising options but require a supine position for harvesting. Numerous other reconstructive options have also been well described, including local and regional flaps to transpositional free flaps. Often, microvascular anastomosis must be performed for flap survival, and these are technically more demanding. In these cases, soft tissue coverage and closure are difficult to obtain. Donor site morbidity and functionality should always be assessed. As with any autograft harvesting, a second incision is required. Given the patient’s age, comorbidities, and functional status, autografts and transfers may be a viable option when indicated. Other autografts are available. Kelahmetoglu and colleagues in 2017 used a free composite anterolateral thigh flap with vascularized fascia lata. Preoperative and postoperative AOFAS scores were 11 and 98, respectively. Visual analog scale of pain decreased from 8 to 1. No complications were reported, and the patient was able to return to daily ambulating activity without support within 5 months. A biomechanical study comparing fascial turndown to dual semitendinosus allograft in terms of tensile strength and construct deformation found that the semitendinosus allograft provided better results [32]. Complications of autografts are donor site morbidity, increased surgical time with risk of infections, graft site dehiscence, graft stretching or elongation.

### 4.6. Allografts

Allografts provide advantages such as reduced donor site complications and surgical time. However, they are subject to risks, even though very low, such as disease transmission and immune response. A 2019 review by Song showed good results with this technique. The mechanical strength of allograft tendon is not inferior compared to autografts [62]. Complications related to this procedure reported in literature are infections, delayed healing of the wound, delayed union of the calcaneal bone block, fragmented calcaneal tuberosity, interosseus ossification and heterotopic bone in the retrocalcaneal bursa [57]. Deese and colleagues retrospectively reviewed 8 patients with deficits ranging from 5.5 cm to 10 cm. All repairs were made with an Achilles tendon allograft with a calcaneal bone block. At final follow-up, no pain or re-ruptures were reported, with good functional outcomes. Minor complications included delayed healing to the incision and heterotopic bone formation in the retrocalcaneal bursa [78]. Other complications reported are immunologic rejection, infection, graft degeneration or weakness, delayed incorporation and risk of disease transmission.

### 4.7. Synthetic Materials and Xenografts

Current data on synthetics and xenografts is mixed. While initial results are encouraging, long-term biological integration and complication rates require further study [66,67,79].

Overall, the most effective techniques depend on the severity of the injury, tissue condition, and availability of tendons to transfer. Postoperative management is crucial to prevent new complications, such as adhesions or infections. Our study focused on Achilles tendon re-rupture. While some studies in the literature primarily address complications related to soft tissues, these were not the focus of our research. It is important to emphasize that in the Achilles tendon region, soft tissue complications often coexist with the underlying tendon. Orthoplasty can usually help the orthopedic surgeon dealing with such comorbidities [19]. Another principle that should be considered when dealing with superficial and deep infected tissues is the obtaining of tissue samples to treat the patient with the most appropriate antibiotic therapy. Additionally, the debridement of infected tissues is essential. Reconstruction techniques must be delayed after the eradication of the infection [80,81,82].

Recent epidemiological evidence has also highlighted the role of endocrine and metabolic pathologies in predisposing to Achilles tendon rupture. In a multicenter study of surgically treated Achilles tendon ruptures, the authors [83] showed that 28% of patients had such metabolic comorbidities, with hypercholesterolemia (42%) and obesity (23%) being the most common, followed by thyroid disorders (18%) and diabetes mellitus (17%). The presence of dysmetabolic diseases significantly increased the likelihood of postoperative medical complications (OR ≈ 1.82) and delayed return to work or sports activities (OR ≈ 0.64). These results emphasize that systemic metabolic dysfunction has a direct and clinically significant impact on tendon healing and surgical outcomes. Therefore, addressing metabolic health may be critical to optimizing recovery from Achilles tendon rupture.

### 4.8. The Role of Artificial Intelligence

Traditional diagnostic and treatment methods rely heavily on clinical examination, imaging techniques, and surgical approaches. The integration of artificial intelligence (AI) into these processes has the potential to significantly improve outcomes in terms of accuracy, precision, and efficiency. AI can augment surgical decision-making by analyzing historical data and patient-specific factors to recommend the most appropriate surgical approach. Machine learning algorithms can analyze large datasets of patient outcomes to identify patterns and predict which surgical techniques are most likely to yield the best results as already reported for other conditions [84,85]. For instance, AI systems can evaluate the success rates of different surgical methods and help surgeons choose the best option based on factors like tendon quality and the patient’s physical condition. Additionally, AI-driven models can be used to assist in preoperative planning. By using patient-specific data, AI can simulate the surgery and predict potential complications, helping the surgical team prepare for challenges that may arise during the procedure. AI is also increasingly integrated into surgical practice through specific devices and tools that may have direct implications for musculoskeletal and tendon surgery. For instance, AI-driven navigation systems such as those developed by Zeta Surgical employ computer vision and real-time tracking to enhance intraoperative precision, with potential applicability to complex tendon revision procedures to improve the reproducibility of reconstruction techniques.

Also, rehabilitation following Achilles tendon repair, particularly after revision surgery, is critical to ensure proper tendon healing and restore function. Kapinski et al. [85] conducted a study aimed at verifying the usefulness of AI in monitoring the post-surgical healing of the Achilles tendon. It can be used to objectively differentiate tissue state and monitoring of the Achilles tendon healing process. Moreover, convolutional neural networks (CNNs) have been successfully applied to ultrasound imaging for the objective assessment of Achilles tendon healing, demonstrating strong correlations with expert radiologist evaluations.

Recovery from an Achilles tendon rupture is often slow and requires a carefully tailored rehabilitation plan that balances rest with progressive loading. AI has the potential to revolutionize rehabilitation by providing personalized, data-driven approaches to recovery, monitoring and tracking patient progress. Wearable devices equipped with AI algorithms can monitor a patient’s activity levels, gait, and mobility in real-time. These devices can assess the biomechanical properties of walking, detect deviations from normal movement patterns, and provide immediate feedback to both the patient and the clinician. Moreover, AI-driven virtual rehabilitation platforms can offer personalized exercises and recovery plans tailored to each patient’s specific needs. These platforms use machine learning to analyze data from wearable devices, feedback from patients, and clinician inputs to adapt rehabilitation protocols. This individualized approach not only accelerates healing but also minimizes the risk of re-injury by ensuring that the patient progresses at an appropriate rate. AI can also enhance the use of robotic systems in rehabilitation, providing precise and controlled movements during physical therapy. Robotic devices can assist with joint mobilization, strengthening exercises, and range-of-motion activities, all while continuously adjusting to the patient’s progress. Kwon et al. [86] designed a wearable paradigm to accurately monitor Achilles tendon load and walking speed using wearable sensors that reduce the burden on the subject and monitor the patient’s load and activity during the recovery phase.

### 4.9. Rehabilitation

Rehabilitation following revision surgery for Achilles tendon rupture requires a tailored, phased approach due to increased complexity and altered tendon integrity. Evidence from randomized controlled trials demonstrates that early controlled weight-bearing and ankle mobilization—initiated within the first one to two postoperative weeks—can enhance the tendon’s healing response, promote metabolic activity (e.g., elevated glutamate and procollagen markers), and improve early range of motion without increasing rupture risk [14,82,87]. In subsequent phases, passive and active range-of-motion exercises and gradual weight-bearing, advancing toward progressive eccentric heel drops, calf raises, and eventually dynamic drills such as single-leg hopping or plyometric training to restore functional biomechanics. Given the complexity of revision surgery, these protocols should be modulated based on tendon quality, comorbidities, and surgical technique, and clearly represent an essential frontier for future research aimed at standardizing outcomes in this challenging patient population.

Rehabilitation following Achilles tendon revision surgery must be adapted to the specific surgical technique employed. When end-to-end repairs or V-Y advancements are performed, early controlled mobilization and progressive eccentric-concentric calf strengthening can usually be introduced sooner. For FHL transfer, early rehabilitation is often more conservative, with gradual progression toward strengthening once tendon integration is secured. Reconstructions using allografts or techniques for defects >6 cm may require extended periods of protected weight-bearing to allow biological incorporation of the graft. The surgical approach also plays a role: endoscopic FHL transfers, compared to open procedures, have been associated with lower complication rates and the potential for earlier initiation of functional exercises, thereby facilitating a faster return to daily and sporting activities. Thus, practical rehabilitation exercises—including calf raises, heel drops, proprioceptive drills, and plyometric activities—must be individualized based on the repair technique, graft choice, and patient-specific tendon healing capacity [88,89,90].

### 4.10. Limitations of the Study

Several limitations of the current study should be noted. First, only articles published in English were considered, which may introduce a potential publication bias. Additionally, despite the use of four major literature databases for the search, the possibility remains that relevant articles could have been missed had other databases been included. Second, the studies included in the analysis varied in terms of evaluation duration, although all studies had a minimum follow-up period of 12 months. It is plausible that both functional outcomes and complication reported may be influenced by the duration of patient follow-up, and these outcomes might differ if a longer or more specific follow-up period were employed. Furthermore, while the search aimed to identify the most effective surgical procedures for the treatment of revision surgery for Achilles tendon rupture, clinicians should not disregard the possibility that differences in patient characteristics could influence the selection of the most appropriate treatment. In this regard, appropriate patient selection and type of Achilles tendon lesions remains a critical factor in optimizing surgical outcomes. Another important limitation is the limited number of specialized centers with the expertise required to perform such complex procedures. This is also reflected in the limited number of available studies and the small patient cohorts reported. The restricted dissemination of this practice contributes to the reduced number of surgeons adequately trained to manage this condition.

However, this review provides an up-to-date overview of revision surgery for Achilles tendon rupture. It offers guidance on surgical decision-making, considering patient condition and lesion type. Emphasis is placed on strategies to optimize functional outcomes. Furthermore, it discusses the potential role of artificial intelligence in supporting diagnosis, treatment planning, and rehabilitation. By integrating clinical evidence with emerging technologies, the review may be of interest to patients, clinicians, and policymakers in the orthopedic field.

## 5. Conclusions

Revision surgery for Achilles tendon injuries still represents a challenge for orthopedic surgeons. The choice of treatment depends on the specific characteristics of the injured tendon and the individual patient’s condition. End-to-end anastomosis is indicated for small tendon defects (<2 cm) with good tendon quality. If the defect is 2 to 5 cm, V-Y myotendinous lengthening should be incorporated while avoiding over-lengthening, which could alter biomechanics and gait. For lesions larger than 5 cm with a viable tendon, fascial turndown is versatile but adds bulk at the repair site, increasing dehiscence risk. The use of tendon transfer, autografts, and allografts, along with lengthening and anastomosis techniques, remains the mainstay of revision surgery for larger defects in a non-viable tendon, while the use of synthetic materials and xenografts is still under study. The integration of artificial intelligence into management has the potential to significantly improve outcomes while monitoring rehabilitation adherence and tendon healing.

## Figures and Tables

**Figure 1 medicina-61-01684-f001:**
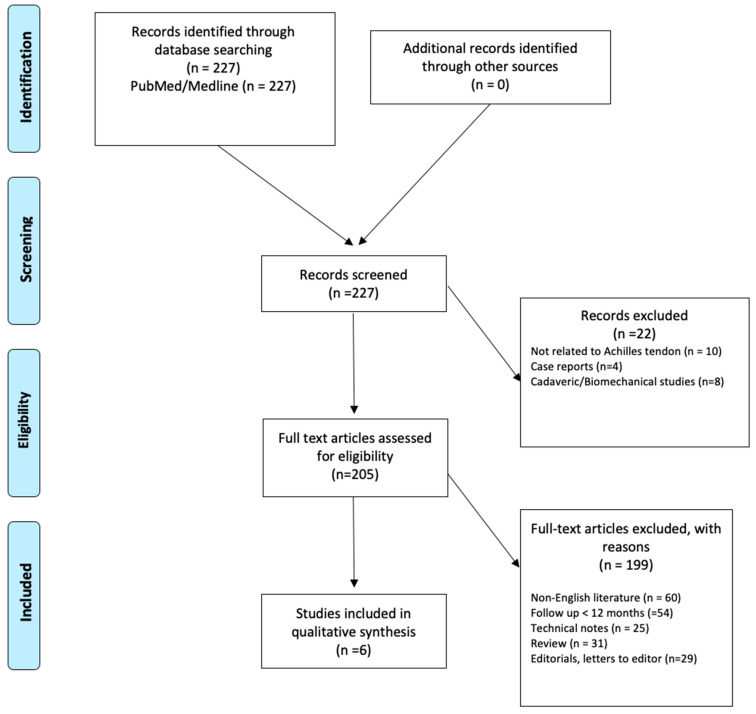
PRISMA flow chart.

**Figure 2 medicina-61-01684-f002:**
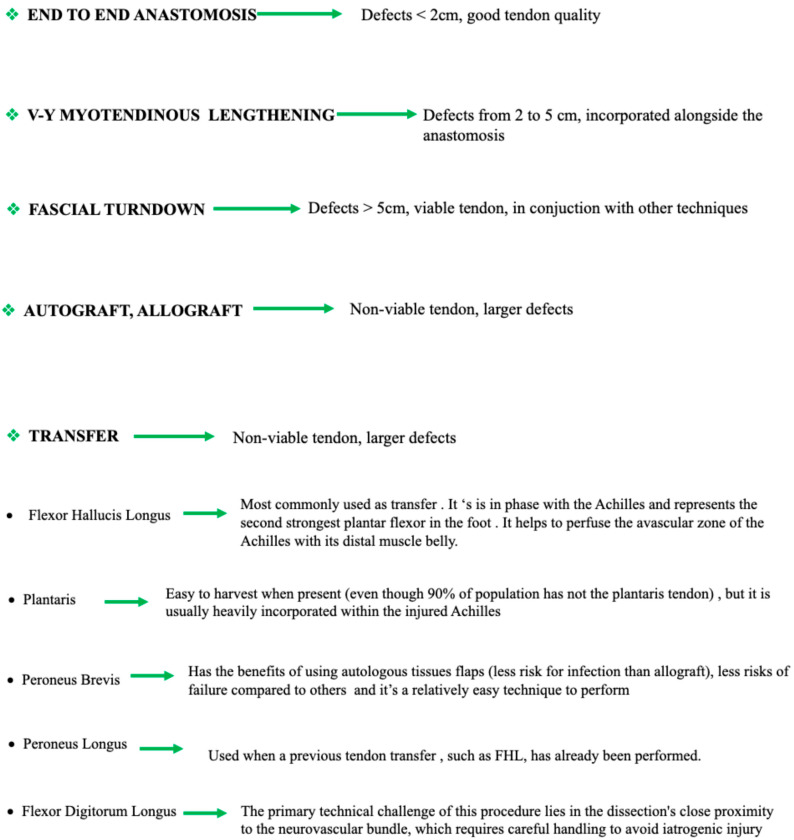
Diagram with indications and surgical techniques.

**Table 1 medicina-61-01684-t001:** Newcastle–Ottawa scale.

Study Author (Year)	Criteria								Total	Quality
	**1**	**2**	**3**	**4**	**5**	**6**	**7**	**8**		
Trivedi et al. (2022) [6]	1	1	1	1	2	1	1	1	9	High
Choi et al. (2024) [7]	1	1	1	1	2	1	1	1	9	High
Khiami et al. (2013) [11]	1	0	1	1	2	1	1	1	8	High
Maffulli et al. (2015) [12]	1	1	1	1	2	1	1	1	9	High
Maffulli et al. (2014) [13]	1	0	1	1	2	1	1	1	8	High
Danford et al. (2023) [14]	1	1	1	1	2	1	1	1	9	High

Based on the total score, quality was classified as “low” (0–3), “moderate” (4–6) and “high” (7–9). Criterion numbers (in bold): 1, representativeness of the exposed cohort; 2, selection of the nonexposed cohort; 3, ascertainment of exposure; 4, demonstration that outcome of interest was not present at start of study; 5, comparability of cohorts on the basis of the design or analysis; 6, assessment of outcome; 7, was follow-up long enough for outcomes to occur?; 8, adequacy of follow-up of cohorts. Each study was awarded a maximum of one or two points for each numbered item within categories, based on the Modified Newcastle-Ottawa scale rules.

**Table 2 medicina-61-01684-t002:** Characteristics of the included studies.

Author	Year	Journal	N.	Sex		Age		Follow-Up	Surgery	Complications	Outcomes
				Male	Female	Mean	SD				
Choi YH et al. [7]	2024	Bone & Joint Research	43,287	33,276	10,011	42.2	11.8	2y	NA	Infection and wound complication (0.5%)	NA
Danford NC et al. [14]	2023	Journal of surgical orthopaedic advances	116	96	20	41.85	14.02	2y	End to end anastomosis open vs minimally invasive	Wound complications (11%), Infection (19%)	Revision surgery rate and complication rate
Maffulli N et al. [13]	2014	Operative Orthopaedics Traumatology	28	21	7	46	9.3	2y	ipsilateral ST transfer	Infection (7.1%)	ATRS (median; ds) 86; 3.63
Maffulli N et al. [12]	2015	International Orthopaedics	21	17	4	38	8.5	3.25y	ipsilateral PB and ST transfer	Weakness calf muscle (19%)	ATRS (median; ds) 82; 3.25
Khiami F et al. [11]	2013	Orthopaedics & Traumatology Surgery & Research	23	20	3	52.1	13	2y	sural triceps aponeurosis transfer	Sural nerve hypoesthesia (0.23%)	AOFAS (median; ds) 96.1; 6.8
Trivedi NN et al. [6]	2023	Sports Health	50,279	34,161	16,118	43.6	11.7	2y	NA	Infection (0.4%), wound complication (0.52%), VTE (1.19%)	Identify risk factors for complications

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
