# Peer review of "Revision Surgery for Achilles Tendon Rupture: A Comprehensive Review of Treatment Options, Outcomes, and Complications and the Role of Artificial Intelligence"

_medicina, 2025, doi:10.3390/medicina61091684_

Round 1
Reviewer 1 Report
Comments and Suggestions for Authors
Review
Many thanks to the authors for having presented so interesting review about “Revision surgery for Achilles tendon rupture: treatment options, outcomes, com-2 plications, and the role of artificial intelligence. A comprehensive review”.
Before resubmitting the revision version of the article, please read the editorial rules carefully, and check other editorial aspects, such as text alignment, text justification at the head, etc. The English language is quite good, generally comprehensible and scientifically structured, but there are several areas where the quality needs improvement for clarity, precision, and academic tone. Correct minor grammar issues, awkward phrasing, and repetitive expressions and improve sentence flow for readability while keeping technical accuracy. Try to use consistent tense and terminology (e.g., “Achilles tendon re-rupture” vs “Achilles re-rupture”). Hence, the manuscript should be corrected by a person of English mother tongue.
General comments
Please, ensuring clear description of study design and eligibility criteria. Strengthening methods transparency (search strategy, inclusion/exclusion criteria, quality assessment. Explicitly describing participants in included studies (age, sex, comorbidities) if available. Clarifying outcome measures and how they were assessed. Making sure limitations and bias risks are explicitly stated and tied to evidence. Expanding interpretation and generalizability of findings.
Detected plagiarism
Detected plagiarism: 10% (it should be < 15%). No direct or obvious plagiarism was detected from this review. Proper references are used, and the structure is consistent with scientific reporting standards.
Title and Abstract
Title identifies the report as a systematic review. Abstract should be fully structured (Background, Objectives, Methods, Results, Conclusions) and state data sources, eligibility criteria, participants, interventions, study appraisal, synthesis methods, main results (including number of included studies), and limitations.
Add explicit mention of the number of studies and participants included in the review and primary outcomes in the abstract results section.
Key words
Please provide them in alphabetic order.
Background
The background and rationale are clearly presented, providing strong clinical rationale. Cites prior research effectively to highlight the gap in literature. Study rationale could be more explicit in identifying the research gap and why this systematic review was needed. Conclude the introduction with a clear, specific research objective or hypothesis, directly matching the PICO framework (Population, Intervention, Comparison, Outcome).
Please, regarding add a few lines regrading Achilles tendon ruptures due to Dysmetabolic Diseases and their management quoting also:
- DOI:10.32098/mltj.03.2018.03
Methods
While databases are named, the exact search strings, date of last search, and any restrictions (language, year) should be reported in full or in a supplement.
NO Protocol and registration: Not stated whether the review was registered in PROSPERO or similar.
Regarding study selection process, it needs a clear description of how many reviewers screened titles/abstracts, how disagreements were resolved, and whether a PRISMA flow diagram was used.
Should describe data items collected, handling of missing data, and whether forms were piloted.
For risk of bias, no explicit tool or method (e.g., Cochrane Risk of Bias tool, ROBINS-I, Newcastle-Ottawa Scale) described.
Meta-analysis methods (if performed) or qualitative synthesis approach not described: wy?
Provide all above details for full PRISMA compliance and the characteristics of included studies’ populations (age, sex, baseline features) should be clearly summarized in Methods/Results.
Statistical analysis
Statistical methods are not sufficiently detailed why?
Results
The results presented are quite complete, reflecting the MM section. They are consistent with methods and clearly presented.
However, provide a PRISMA flow diagram showing number of records identified, screened, excluded (with reasons), and included.
Tables summarizing included studies should include author/year, country, sample size, participant characteristics, intervention, comparator, follow-up duration, and outcomes.
The results synthesis lacks structured presentation consider splitting into primary and secondary outcomes, reporting effect estimates with confidence intervals, and heterogeneity measures if applicable.
The outcome data not consistently stratified by important confounders (sex, age, comorbidities) and missing clarity on follow-up periods and loss to follow-up in included studies.
Discussion
The length and content of the discussion communicates the main information of the paper. The discussion section interprets findings in the context of national trends but could more thoroughly integrate literature to contextualize results. Integrates findings with prior literature. List more limitations but should specify the direction and magnitude of potential bias.
The authors should improve: comparison with other reviews and high-quality primary studies.
Limitations section should explicitly address: risk of bias in included studies; limitations of the search strategy (databases, language restrictions); lack of high-quality RCTs or observational studies.
Finally, the implications for clinical practice and research should be more concrete.
Conclusions
The conclusions provide a clear summary of the main points of the review, and they only reflect and refer to its results.
References
The references are up to date, but they should be integrated as suggested previously.
Figures and Tables:
Ensure captions are self-contained, all abbreviations used in figures are defined.
Competing interest
There are no competing interests. Conflicts of interest: Should be explicitly stated for all authors.
Concerns
The paper does not raise any concerns (no self-citations probably).
Funding and Conflicts of Interest
Not stated in the visible parts of the manuscript. Hence, add a section disclosing funding sources and potential conflicts of interest and PRISMA requires reporting funding sources for the review and for included studies if available.
Comments on the Quality of English Language
Correct minor grammar issues, awkward phrasing, and repetitive expressions and improve sentence flow for readability while keeping technical accuracy. Try to use consistent tense and terminology (e.g., “Achilles tendon re-rupture” vs “Achilles re-rupture”). Hence, the manuscript should be corrected by a person of English mother tongue.
Author Response
Reviewer #1
Review Many thanks to the authors for having presented so interesting review about “Revision surgery for Achilles tendon rupture: treatment options, outcomes, complications, and the role of artificial intelligence. A comprehensive review”.
Before resubmitting the revision version of the article, please read the editorial rules carefully, and check other editorial aspects, such as text alignment, text justification at the head, etc. The English language is quite good, generally comprehensible and scientifically structured, but there are several areas where the quality needs improvement for clarity, precision, and academic tone. Correct minor grammar issues, awkward phrasing, and repetitive expressions and improve sentence flow for readability while keeping technical accuracy. Try to use consistent tense and terminology (e.g., “Achilles tendon re-rupture” vs “Achilles re-rupture”). Hence, the manuscript should be corrected by a person of English mother tongue.
A: Thank you for your comment. The English language and terminology have been carefully corrected.
General comments Please, ensuring clear description of study design and eligibility criteria. Strengthening methods transparency (search strategy, inclusion/exclusion criteria, quality assessment. Explicitly describing participants in included studies (age, sex, comorbidities) if available. Clarifying outcome measures and how they were assessed. Making sure limitations and bias risks are explicitly stated and tied to evidence. Expanding interpretation and generalizability of findings.
A: Thank you for your comment. We revised the manuscript accordingly. The study specifies the inclusion and exclusion criteria, the search strategy, and the quality assessment conducted using the Newcastle-Ottawa Scale. The Table 2 was expanded to include detailed information on patients’ age, sex, follow up, comorbidities, and outcomes, with the aim of enhancing the clarity and visual presentation of the collected data. We also revised the discussion and we clarified limits of our review.
Detected plagiarism: 10% (it should be < 15%). No direct or obvious plagiarism was detected from this review. Proper references are used, and the structure is consistent with scientific reporting standards.
A: Thank you. We appreciate.
Title and Abstract Title identifies the report as a systematic review. Abstract should be fully structured (Background, Objectives, Methods, Results, Conclusions) and state data sources, eligibility criteria, participants, interventions, study appraisal, synthesis methods, main results (including number of included studies), and limitations.
Add explicit mention of the number of studies and participants included in the review and primary outcomes in the abstract results section.
A: Thank you for your valuable suggestion. We have implemented the suggested revisions in the Abstract section.
Key words Please provide them in alphabetic order.
A: Thank you. We made it
Background The background and rationale are clearly presented, providing strong clinical rationale. Cites prior research effectively to highlight the gap in literature. Study rationale could be more explicit in identifying the research gap and why this systematic review was needed. Conclude the introduction with a clear, specific research objective or hypothesis, directly matching the PICO framework (Population, Intervention, Comparison, Outcome).
Please, regarding add a few lines regrading Achilles tendon ruptures due to Dysmetabolic Diseases and their management quoting also:
- DOI:10.32098/mltj.03.2018.03
A: Thank you for your comment. We have clarified the study outcomes that distinguish it compared to other works in the literature. As suggested, we have included a brief section on metabolic disorders contributing to Achilles tendon rupture, citing the relevant article indicated.
Methods While databases are named, the exact search strings, date of last search, and any restrictions (language, year) should be reported in full or in a supplement.
NO Protocol and registration: Not stated whether the review was registered in PROSPERO or similar.
Regarding study selection process, it needs a clear description of how many reviewers screened titles/abstracts, how disagreements were resolved, and whether a PRISMA flow diagram was used.
Should describe data items collected, handling of missing data, and whether forms were piloted.
For risk of bias, no explicit tool or method (e.g., Cochrane Risk of Bias tool, ROBINS-I, Newcastle-Ottawa Scale) described.
Meta-analysis methods (if performed) or qualitative synthesis approach not described: wy?
Provide all above details for full PRISMA compliance and the characteristics of included studies’ populations (age, sex, baseline features) should be clearly summarized in Methods/Results.
A: Thank you for the suggestion. We have clarified and implemented the requested details regarding the search, PRISMA guidelines, PICO framework, reviewers’ activities, and risk of bias assessment as suggested. The study was not registered on PROSPERO, and no meta-analyses were performed as the study was intended as comprehensive review. Additionally, we have expanded the Results section and revised the Tables with other useful information as suggested.
Statistical analysis Statistical methods are not sufficiently detailed why?
A:Thank you for the suggestion. As this is a comprehensive review, no statistical analyses were performed. Furthermore, the extracted data is heterogeneous and limited, not allowing for aggregation for statistical purposes.
Results The results presented are quite complete, reflecting the MM section. They are consistent with methods and clearly presented.
However, provide a PRISMA flow diagram showing number of records identified, screened, excluded (with reasons), and included.
Tables summarizing included studies should include author/year, country, sample size, participant characteristics, intervention, comparator, follow-up duration, and outcomes.
The results synthesis lacks structured presentation consider splitting into primary and secondary outcomes, reporting effect estimates with confidence intervals, and heterogeneity measures if applicable.
The outcome data not consistently stratified by important confounders (sex, age, comorbidities) and missing clarity on follow-up periods and loss to follow-up in included studies.
A: We appreciate your comment. We revised the PRISMA flow diagram and expanded Table 2 to include additional relevant information extracted from the reviewed articles as suggested. Furthermore, we have clarified the definition of the primary and secondary outcomes
Discussion The length and content of the discussion communicates the main information of the paper. The discussion section interprets findings in the context of national trends but could more thoroughly integrate literature to contextualize results. Integrates findings with prior literature. List more limitations but should specify the direction and magnitude of potential bias.
The authors should improve: comparison with other reviews and high-quality primary studies.
Limitations section should explicitly address: risk of bias in included studies; limitations of the search strategy (databases, language restrictions); lack of high-quality RCTs or observational studies.
Finally, the implications for clinical practice and research should be more concrete.
A: We appreciate your suggestion. In response, we have expanded the discussion section to integrate literature to contextualize results and added additional references, including those providing insights into rehabilitation. Furthermore, we revised the limitations section to more clearly delineate the study’s scope, constraints, and generalizability.
Conclusions The conclusions provide a clear summary of the main points of the review, and they only reflect and refer to its results.
A: Thank you for your comment.
References
Figures and Tables: Ensure captions are self-contained, all abbreviations used in figures are defined.
A: Thank you.
Competing interest There are no competing interests. Conflicts of interest: Should be explicitly stated for all authors.
A: Thank you. We specify no conflict for all the authors at the final pages
Concerns The paper does not raise any concerns (no self-citations probably).
A: Thank you. We appreciate.
Funding and Conflicts of Interest Not stated in the visible parts of the manuscript. Hence, add a section disclosing funding sources and potential conflicts of interest and PRISMA requires reporting funding sources for the review and for included studies if available.
A: Thank you. We specify no conflict for all the authors at the final pages
Reviewer 2 Report
Comments and Suggestions for Authors
The manuscript presents a comprehensive review of revision surgery for Achilles tendon ruptures, focusing on indications, surgical techniques, outcomes, and complications. The authors stratify surgical strategies by defect size (<2 cm, 2–5 cm, >5 cm) and discuss a range of options including end-to-end anastomosis, V–Y myotendinous lengthening, fascial turndown flaps, tendon transfers, autografts, allografts, and synthetic materials. The review also dedicates a section to the potential applications of artificial intelligence in preoperative planning, complication prediction, and rehabilitation monitoring. The work aims to assist orthopedic surgeons in selecting patient-specific, defect-oriented treatment strategies and to highlight emerging technologies for improving outcomes.
While the review covers a breadth of techniques and introduces an emerging technology dimension, several critical revisions are needed to meet academic and clinical impact standards:
- The current manuscript does not clearly articulate how it differs from prior systematic or narrative reviews on revision Achilles tendon surgery. A dedicated subsection in the Introduction or Discussion should explicitly identify recent related reviews, highlight the gaps they leave, and explain how the present paper fills those gaps. Without this, the reader may struggle to understand the added value of the work.
- Several references are incomplete or missing essential bibliographic details.
Example 1: Ref. 2 currently reads:
“Huttunen, T.T.; Kannus, P.; Rolf, C.; Felländer-Tsai, L.; Mattila, V.M. Acute Achilles tendon ruptures. Am. J. Sports Med. 2014, 42, 2419–2423.”
This omits the full title:
Huttunen, T. T., Kannus, P., Rolf, C., Felländer-Tsai, L., & Mattila, V. M. (2014). Acute Achilles tendon ruptures: Incidence of injury and surgery in Sweden between 2001 and 2012. The American Journal of Sports Medicine, 42(10), 2419–2423.
Example 2: Ref. 57 lacks issue number and should read:
Aktas, S., & Kocaoglu, B. (2009). Open versus minimal invasive repair with Achillon device. Foot & Ankle International, 30(5), 391–397.
Example 3: Ref. 74 matches Kuwada’s 1990 paper but the title in the reference list differs from the original. The correct form is:
Kuwada, G. T. (1990). Classification of tendo Achillis rupture with consideration of surgical repair techniques. The Journal of Foot Surgery, 29(4), 361–365.
A thorough audit of all references is necessary to ensure full compliance with MDPI and journal style requirements.
- The review would benefit from summary tables comparing the main revision techniques in terms of:
- Indications (defect size, tissue quality)
- Functional outcomes (AOFAS, return-to-sport rates)
- Complication rates
- Donor site morbidity (for autografts)
- Advantages/disadvantages
Including algorithmic flowcharts or decision trees adapted from Myerson/Buda and modified with current evidence would enhance clinical usability.
- The AI section is promising but remains generic. To strengthen it:
- Provide specific examples of existing AI tools or machine learning models applied in musculoskeletal surgery or tendon repair.
- Discuss validation studies, regulatory considerations, and implementation barriers in orthopedic settings.
- Integrate AI concepts into the surgical discussion—e.g., how predictive analytics might influence the choice between FHL transfer vs. allograft in a specific patient profile.
- Indicate whether any quality assessment beyond the Modified Newcastle–Ottawa Scale was performed (e.g., GRADE for evidence strength).
- Clarify whether the included six studies specifically addressed revision surgery or if some also included primary repairs, and how data extraction handled such cases.
- Structure and Presentation:
- The PRISMA methodology is described but lacks a detailed table of excluded studies and reasons for exclusion, which would strengthen transparency.
- Figures and tables should be cross-referenced in the text with specific explanatory commentary rather than left to stand alone.
- Perform a global search to replace all instances of parenthetical reference numbers (n) or (n,n) with square-bracketed references [n] and ensure they precede the final punctuation.
- Confirm all references in text match the numbering and order in the reference list after correction.
- Maintain consistency even for single citations (e.g., [15] instead of (15)).
- Minor typographical and grammatical inconsistencies should be corrected (e.g., spacing before unit symbols, uniform use of decimal points, consistent capitalization of surgical terms).
Author Response
Reviewer #2
While the review covers a breadth of techniques and introduces an emerging technology dimension, several critical revisions are needed to meet academic and clinical impact standards:
- The current manuscript does not clearly articulate how it differs from prior systematic or narrative reviews on revision Achilles tendon surgery. A dedicated subsection in the Introduction or Discussion should explicitly identify recent related reviews, highlight the gaps they leave, and explain how the present paper fills those gaps. Without this, the reader may struggle to understand the added value of the work.
A: Thank you for the comment. We clarified the study outcomes that distinguish and identify it compared to other works in the literature at the end of the discussion
- Several references are incomplete or missing essential bibliographic details.
A: Thank you for the comment, they were corrected.
- The review would benefit from summary tables comparing the main revision techniques in terms of:
- Indications (defect size, tissue quality)
- Functional outcomes (AOFAS, return-to-sport rates)
- Complication rates
- Donor site morbidity (for autografts)
- Advantages/disadvantages
Including algorithmic flowcharts or decision trees adapted from Myerson/Buda and modified with current evidence would enhance clinical usability.
A: Thank you for the comment. Table 2 and Figure 2 and text have been updated to include the requested parameters, as far as they could be extracted from the included scientific studies.
- The AI section is promising but remains generic. To strengthen it:
- Provide specific examples of existing AI tools or machine learning models applied in musculoskeletal surgery or tendon repair.
- Discuss validation studies, regulatory considerations, and implementation barriers in orthopedic settings.
- Integrate AI concepts into the surgical discussion—e.g., how predictive analytics might influence the choice between FHL transfer vs. allograft in a specific patient profile.
A: Thank you for the comment. We improve the AI section with the suggestions indicated.
- Indicate whether any quality assessment beyond the Modified Newcastle–Ottawa Scale was performed (e.g., GRADE for evidence strength).
A: Thank you for the comment. We performed only Newcastle Ottawa scale for the quality assessment.
- Clarify whether the included six studies specifically addressed revision surgery or if some also included primary repairs, and how data extraction handled such cases.
A: Thank you for the comment. The six studies addressed only cases of revision surgery. We considered studies on primary repairs as exclusion criteria to reduce bias.
- Structure and Presentation:
- The PRISMA methodology is described but lacks a detailed table of excluded studies and reasons for exclusion, which would strengthen transparency.
- Figures and tables should be cross-referenced in the text with specific explanatory commentary rather than left to stand alone.
- Perform a global search to replace all instances of parenthetical reference numbers (n) or (n,n) with square-bracketed references [n] and ensure they precede the final punctuation.
- Confirm all references in text match the numbering and order in the reference list after correction.
- Maintain consistency even for single citations (e.g., [15] instead of (15)).
- Minor typographical and grammatical inconsistencies should be corrected (e.g., spacing before unit symbols, uniform use of decimal points, consistent capitalization of surgical terms).
A: Thank you for the comment. All the issues raised have been appropriately addressed
Reviewer 3 Report
Comments and Suggestions for Authors
Dear Authors,
Thank you for submitting this article. I appreciate the theme and the fact that you aim to clear the management protocol of Achilles tendon revision surgery procedures.
Revision surgery in any field of orthopaedics is a challenge and requires, both from the surgeon and the patient, a lot of knowledge and discipline to obtain the best results.
In the treatment algorithm I think is necessary to better define the underlying pathologies, the co-morbities and the healing potential.
In the section 3.1.4 and in the discussion section I consider the need to address the arthroscopic FHL transfer as it has been shown that it can provide a reliable solution in revision cases, without the major complications associated with open surgery.
I also think that the rehab protocol differences can lead to different results. So I would like you to take into consideration some paragraphs on this matter. There is also a need to better detail the outcomes section, which I cannot se in the discussion phase although it is mentioned in the title.
The artificial intelligence section from my point of view has many general ideas, but not quite focused on the proposed theme. There are only 2 studies mentioned and it not very clearly defined how AI has enhanced the treatment algorithm.
Thank you!
Author Response
Reviewer #3
Dear Authors,
Thank you for submitting this article. I appreciate the theme and the fact that you aim to clear the management protocol of Achilles tendon revision surgery procedures.
Revision surgery in any field of orthopaedics is a challenge and requires, both from the surgeon and the patient, a lot of knowledge and discipline to obtain the best results.
In the treatment algorithm I think is necessary to better define the underlying pathologies, the co-morbities and the healing potential.
In the section 3.1.4 and in the discussion section I consider the need to address the arthroscopic FHL transfer as it has been shown that it can provide a reliable solution in revision cases, without the major complications associated with open surgery.
I also think that the rehab protocol differences can lead to different results. So I would like you to take into consideration some paragraphs on this matter. There is also a need to better detail the outcomes section, which I cannot se in the discussion phase although it is mentioned in the title.
The artificial intelligence section from my point of view has many general ideas, but not quite focused on the proposed theme. There are only 2 studies mentioned, and it not very clearly defined how AI has enhanced the treatment algorithm.
Thank you!
A: We appreciate your suggestion. We have improved the therapeutic algorithm section and the 3.1.4 section with the elements you requested.
We have also made a new paragraph on rehabilitation as you suggested.
We also improved the AI section with new further specific information.
Round 2
Reviewer 1 Report
Comments and Suggestions for Authors
I would like to express my sincere gratitude to the authors for taking my comments into consideration.
However, in the Discussion section, it would greatly enhance the clarity and scientific robustness of the manuscript to better articulate the role of endocrine and metabolic pathologies, such as diabetes mellitus, hypercholesterolemia, thyroid disorders, and obesity in impairing tendon health and predisposing to Achilles tendinopathy and rupture. A recent multicentric epidemiologic study has reported that these metabolic conditions compromise tendon integrity and are likely predisposing factors for Achilles tendon ruptures. Please, discuss this quoting: https://pubmed.ncbi.nlm.nih.gov/35806982/
Reviewer 2 Report
Comments and Suggestions for Authors
The authors have addressed the main concerns raised in the initial review. The revised manuscript now presents a clearer, more rigorous, and clinically useful synthesis of revision surgery for Achilles tendon rupture, with the added perspective of artificial intelligence applications.
The paper is substantially improved and can be considered acceptable after minor editorial corrections to reference style and language.
Author Response
Thank you for the comment